# Age-related demethylation of the TDP-43 autoregulatory region in the human motor cortex

Yuka Koike [1], Akihiro Sugai [2✉], Norikazu Hara[3], Junko Ito[4], Akio Yokoseki[5], Tomohiko Ishihara[1], Takuma Yamagishi[1], Shintaro Tsuboguchi[1], Mari Tada[6], Takeshi Ikeuchi [3], Akiyoshi Kakita[4] & Osamu Onodera [1✉]

In amyotrophic lateral sclerosis (ALS), TAR DNA-binding protein 43 (TDP-43), which is encoded by *TARDBP*, forms aggregates in the motor cortex. This aggregate formation may be triggered by an increase in the TDP-43 level with aging. However, the amount of TDP-43 is autoregulated by alternative splicing of the *TARDBP* 3′UTR, and how this autoregulation is affected by aging remains to be elucidated. We found that DNA demethylation in the autoregulatory region in the *TARDBP* 3′UTR reduced alternative splicing and increased *TARDBP* mRNA expression. Furthermore, in the human motor cortex, we found that this region was demethylated with aging, resulting in increased expression of *TARDBP* mRNA. The acceleration of DNA demethylation in the motor cortex was associated with the age of ALS onset. In summary, the dysregulation of TDP-43 autoregulation by age-related DNA demethylation in the motor cortex may explain the contribution of aging and motor system selectivity in ALS.

[1] Department of Neurology, Brain Research Institute, Niigata University, Niigata city, Japan. [2] Department of Molecular Neuroscience, Center for Bioresource-based Research, Brain Research Institute, Niigata University, Niigata city, Japan. [3] Department of Molecular Genetics, Center for Bioresource-based Research, Brain Research Institute, Niigata University, Niigata city, Japan. [4] Department of Pathology, Brain Research Institute, Niigata University, Niigata city, Japan. [5] Department of Inter-Organ Communication Research, Niigata University Graduate School of Medical and Dental Sciences, Niigata city, Japan. [6] Department of Pathology Neuroscience, Center for Bioresource-based Research, Brain Research Institute, Niigata University, Niigata city, Japan. ✉email: akihiro.sugai@bri.niigata-u.ac.jp; onodera@bri.niigata-u.ac.jp

In neurodegenerative diseases with aging as a primary risk factor[1], specific proteins aggregate selectively in the central nervous system. It has been speculated that an increase in the amount of causative proteins contributes to aggregate formation[2,3]. Thus, one pathological hypothesis of system selectivity in neurodegenerative diseases is that the amount of disease-causing protein is increased in the impaired system. In fact, alpha-synuclein levels have been shown to be increased in neuronal systems affected by alpha-synucleinopathy[4]. However, the association between the amount of the causative protein and aging is unclear.

Sporadic amyotrophic lateral sclerosis (ALS) is a neurodegenerative disease that selectively impairs primary motor neurons in the motor cortex and alpha-motor neurons in the spinal cord, and similar to other sporadic neurodegenerative diseases, aging is a major risk factor[5]. In more than 95% of sporadic ALS cases, the nuclear protein TDP-43 forms aggregates in the cytoplasm and is depleted from the nucleus in diseased cells[6,7,8]. Increased intracellular concentrations of TAR DNA-binding protein 43 (TDP-43) have been speculated to be a factor in this aggregate formation[9,10]. However, to date, the mechanism linking TDP-43, motor neuron selectivity, and aging has not been elucidated.

Nuclear TDP-43 binds the *TARDBP* pre-mRNA 3′UTR, and the level of TDP-43 is strictly autoregulated via its alternative splicing[11–13]. An increased level of nuclear TDP-43 promotes its splicing to produce nonsense-mediated mRNA decay-sensitive *TARDBP* mRNA and reduce the level of TDP-43[12,14]. However, when the level of TDP-43 in the nucleus is reduced, this splicing is repressed, and the *TARDBP* mRNA levels are increased. Therefore, *TARDBP* mRNA expression is continually increased in ALS-affected cells with reduced nuclear TDP-43 levels[14]. Following the formation of cytoplasmic TDP-43 aggregates, which are presumed to reduce the TDP-43 level in the nucleus by either entrapping newly generated TDP-43 or inhibiting the nuclear transport of TDP-43, the progression of the disease accelerates[15,16]. Some mutations in the *TARDBP* gene can alter its alternative splicing and affect the autoregulatory mechanism[17,18]. However, in sporadic ALS, the factor triggering the disruption of this autoregulatory mechanism leading to increased TDP-43 expression and TDP-43 aggregate formation in the motor cortex is unclear.

Aging alters the methylation status of DNA. The DNA methylation status defines the epigenetic age independent of the chronological age. In age-related diseases, this epigenetic age contributes more than the chronological age to mortality and morbidity[19]. Indeed, an accelerated epigenetic age based on DNA methylation is associated with the age of onset in ALS patients with *C9orf72* expansion repeats[20] and in sporadic ALS patients[21]. DNA methylation not only regulates gene expression but also affects alternative splicing[22–26]. Based on these observations, we hypothesized that the DNA methylation status of the *TARDBP*-3′ UTR changes with age in the motor cortex, which in turn alters the splicing of the *TARDBP*-3′UTR in the direction of increased TDP-43 expression.

To explore this hypothesis, we used the dCas9 system to selectively demethylate the CpG region of the alternative splicing-related site in the *TARDBP* 3′UTR. We found that the demethylation of this region inhibited the alternative splicing and increased the expression of *TARDBP* mRNA. Furthermore, in the human motor cortex, this region was demethylated with aging, and the degree of demethylation was correlated with the level of *TARDBP* mRNA expression. An accelerated DNA methylation age of the *TARDBP* 3′UTR in the ALS motor cortex was associated with a younger age of onset, suggesting its involvement in ALS pathogenesis.

## Results

**CpG sites are clustered around the alternative splicing site in the *TARDBP* 3′UTR.** TDP-43 binds the 3′UTR of *TARDBP* pre-mRNA and induces the alternative splicing of intron 6 (exonic intron) and intron 7 (3′UTR intron), leading to nonsense-mediated mRNA decay[12,14]. We found that 15 CpG sites were clustered around the 5′ splicing site of alternative intron 7. An analysis of publicly available data revealed that these CpG sites are moderately to highly methylated in the human prefrontal cortex (Fig. 1a)[27,28]. At methylated DNA sites, RNA polymerase

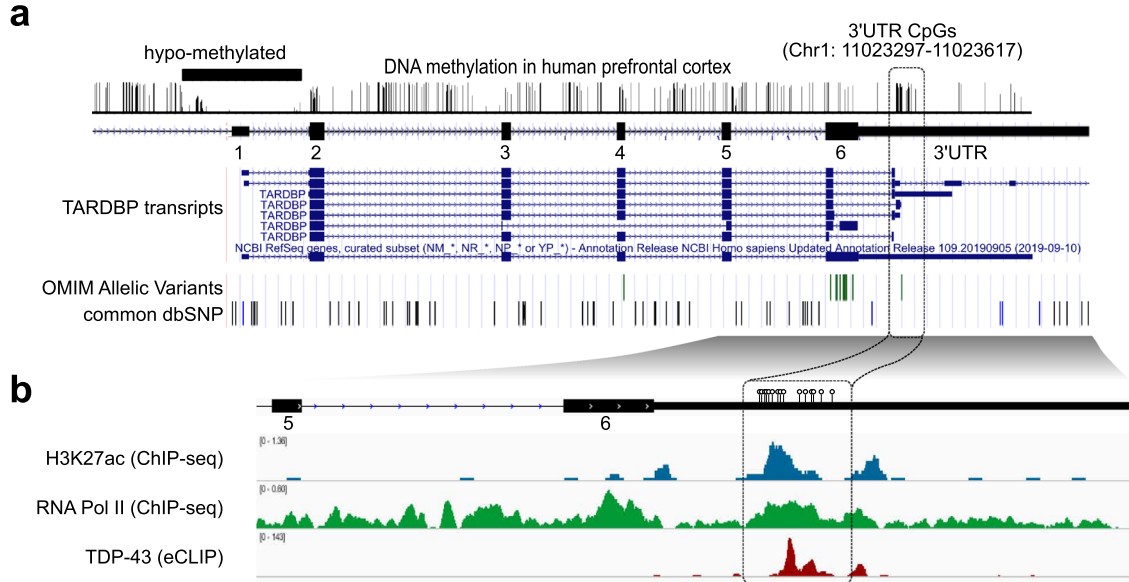

**Fig. 1 *TARDBP* 3′UTR CpG sites. a** The *TARDBP* DNA methylation status in the human prefrontal cortex obtained from MethBase. *TARDBP* transcripts, ALS-causing gene variants (OMIM allelic variants), and SNP positions as indicated in the Single Nucleotide Polymorphism Database (dbSNP) were determined with the UCSC Genome Browser on Human Dec. 2013 (GRCh38/hg38) Assembly. **b** CpG (circles) located around the 5′ site of intron 7 in magnified views of exons 5, 6 and 3′UTR, ChIP-seq data of H3K27ac (SRX2161995) and RNA polymerase II (SRX100455) in K562 cells as indicated by ChIP-Atlas and eCLIP data of TDP-43 in K562 cells from the ENCODE database (ENCSR584TCR).

II may stall, eliciting splicing[23,24,29]. By analyzing publicly available data, we found that the CpG cluster region in the *TARDBP* 3′ UTR is enriched in RNA polymerase II[30] (Fig. 1b), which may be associated with the alternative splicing of intron 7[31]. We found that this region is also enriched in H3K27ac marks (Fig. 1b)[30], which is a characteristic of regions where DNA methylation changes with age[32,33]. Thus, we hypothesized that the mRNA level of *TARDBP* increases with age via DNA demethylation in this region.

**Demethylation of the *TARDBP* 3′UTR suppresses alternative splicing and increases *TARDBP* mRNA expression.** To test our hypothesis, we first examined whether demethylation of this region suppressed splicing. To manipulate the DNA methylation status at these 15 CpG sites, we applied the dCas9 system with four guide RNAs (Fig. 2a). The TET1-*TARDBP*-target vector[34] or the DNMT3A-*TARDBP*-target vector[35] was applied for demethylation or methylation, respectively, at the 15 CpG sites (Fig. 2b).

CpGs 1–15 in the 3′UTR were highly methylated in HEK293T cells (Fig. 2c; control). The transfection of the TET1-*TARDBP*-target vector with each guide RNA demethylated these CpG sites (Fig. 2c). The 3′UTR CpG 10–15 sites were more efficiently demethylated than the 3′UTR CpG 1–9 sites, which were negatively correlated with the CpG density (Fig. 2c–e, Supplementary Fig. 1a, b). However, the transfection of the DNMT3A-*TARDBP*-target vector did not alter the DNA methylation percentage (Fig. 2c).

Then, we investigated whether the heterogeneity of each epiallele was altered upon region-specific TET1-dependent demethylation, as age-related DNA methylation changes have been reported to increase heterogeneity in DNA methylation[36] and decrease the correlation between the methylation status of each CpG pair[37]. The methylation patterns of cells transfected with the TET1-*TARDBP*-target vector showed a higher epipolymorphism score than that of control cells at CpGs 10–15 (Fig. 2f–h, Supplementary Fig. 1c, d)[38,39]. Furthermore, the methylation-linkage diagram revealed that the correlation of the

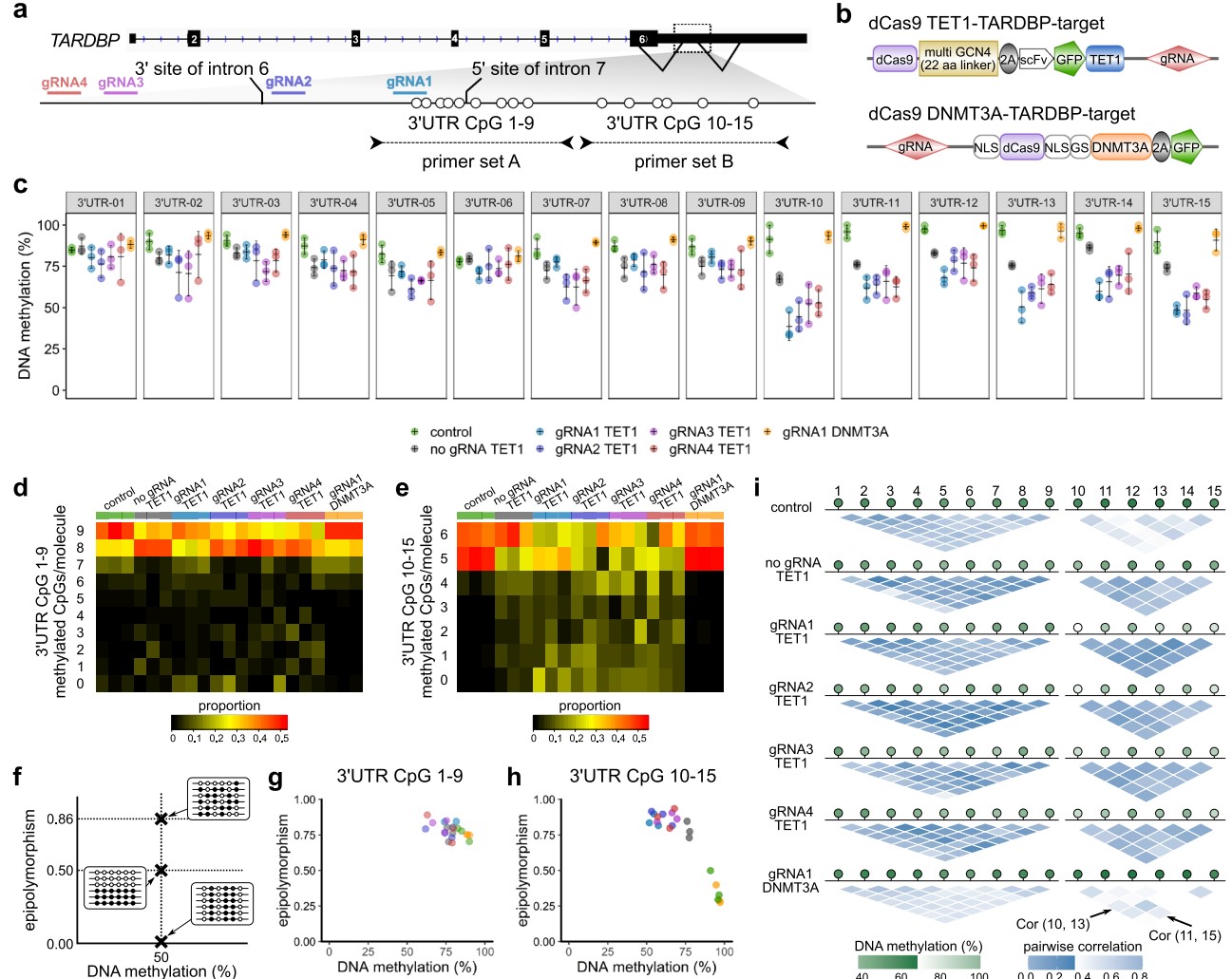

**Fig. 2 Targeted manipulation of the DNA methylation status of the *TARDBP* 3′UTR. a** Schematic of the CpG sites of the *TARDBP* gene, the target regions of the guide RNAs (gRNA1, 2, 3, and 4), and primer sets A and B used for the bisulfite amplicon sequencing. **b** Structure of the vectors used in the experiment. **c** Percent DNA methylation at 3′UTR CpGs 1–15 in HEK293T cells (mean ± SD, n = 3). **d, e** Proportions of the number of sequence reads classified by the number of CpGs methylated among nine CpG sites (3′UTR CpGs 1–9) or six CpG sites (3′UTR CpGs 10–15). **f** Schematic of epipolymorphisms showing methylation profiles with the same mean methylation level but different methylation patterns within the CpG region. Black and white circles represent methylated CpGs and demethylated CpGs, respectively. **g, h** Scatterplot showing the epipolymorphism score of the DNA methylation percentage. **i** Methylation linkage diagram of 3′UTR CpGs 1–9 and 3′UTR CpGs 10–15 showing the correlation of all CpG-site pairs in the target region. Green circles indicate the mean percentage of DNA methylation at each CpG site.

methylation status of each CpG pair was increased by TET1 (Fig. 2i). These results indicate that TET1 affects each CpG on the allele as a cluster rather than stochastically.

Then, we examined whether demethylation in the region suppresses the alternative splicing of *TARDBP* mRNA (Fig. 3a). In the cells transfected with the TET1-*TARDBP*-target vector with guide RNA1, the alternative splicing of intron 6 and intron 7 was suppressed (Fig. 3b, c) with a concomitant 1.85-fold increase in the level of unspliced canonical mRNA (Fig. 3d). These results demonstrate that the demethylation of 3′UTR CpGs 10–15, which are sensitive to TET1, suppresses the alternative splicing of intron 6 and intron 7 and increases the *TARDBP* mRNA levels (Fig. 3e).

**DNA methylation of *TARDBP* in the human brain**. Subsequently, we explored the DNA methylation status of the *TARDBP* gene in the human brain by examining the motor, occipital, and cerebellar cortexes of seven patients with sporadic ALS and eight control patients without brain disease (Supplementary Table 1: case; ALS 1–7, control 1–8). We investigated 22 CpG sites in the *TARDBP* promoter region that were predicted by PROMOTER SCAN[40] and PromoterInspector[41] and the 15 CpG sites in the *TARDBP* 3′UTR (Fig. 4a).

In the examined brain regions, the CpG sites in the promoter region were hypomethylated, while 3′UTR CpGs 1–15 were moderately to highly methylated (Fig. 4b–j). The percentage of DNA methylation in 3′UTR CpGs 10–15 was lower than that in 3′UTR CpGs 1–9 (Fig. 4, Supplementary Fig. 2a) and was positively correlated with the CpG density (Supplementary Fig. 2b). The interindividual variability in the percentage of methylation in 3′UTR CpGs 10–15 was greater than that in 3′UTR CpGs 1–9 (Fig. 4, Supplementary Fig. 2c) and was also associated with the surrounding CpG density (Supplementary Fig. 2d)[42]. The cerebral cortex exhibited a lower percentage of 3′UTR CpG 10–15 methylation and greater interindividual

variability than the cerebellum (Fig. 4, Supplementary Fig. 2c). In these brain regions, the percentage of DNA methylation at each CpG did not differ between the ALS patients and the controls (Fig. 4b–j).

**Characteristics of *TARDBP* 3′UTR DNA methylation in the motor cortex**. To evaluate the tissue- and age-related methylation patterns in the motor cortex, we further examined the motor cortex of three ALS patients and three controls, resulting in a total of 10 ALS patients and 11 controls (Supplementary Table 1). As a noncentral nervous system tissue, we analyzed DNA obtained from the livers of ALS patients. Among 3′UTR CpGs 1–9, the motor cortex exhibited more epialleles with demethylated CpGs than the other regions (Fig. 5a) with high pairwise correlations in each CpG site (Fig. 5c). In 3′UTR CpGs 10–15, the motor and occipital cortex presented more epialleles with demethylated CpGs (Fig. 5b) with high pairwise correlations in each CpG site (Fig. 5c) compared to those in the cerebellum. The epipolymorphism scores of the nervous tissues were higher than those of the liver (Supplementary Fig. 3a–f). These features were similar in the ALS and control groups. Each tissue was classified by a principal component analysis incorporating these indices (Fig. 5d). The motor cortex was also characterized by high interindividual variability (Fig. 5d).

**Age-related demethylation of the *TARDBP* 3′UTR in the motor cortex**. Subsequently, we investigated the association between the DNA methylation status of the 3′UTR CpGs in the motor cortex and age (Fig. 6a–c). In the control motor cortex, we found that the percentage of DNA methylation at 3′UTR CpGs 10–15 was inversely correlated with age (Fig. 6a, c). The percentage of DNA methylation was approximately 80% in the individuals in their 50 s but decreased to nearly 50% in the individuals in their 80 s. Among the elderly, the proportion of epialleles with five or more

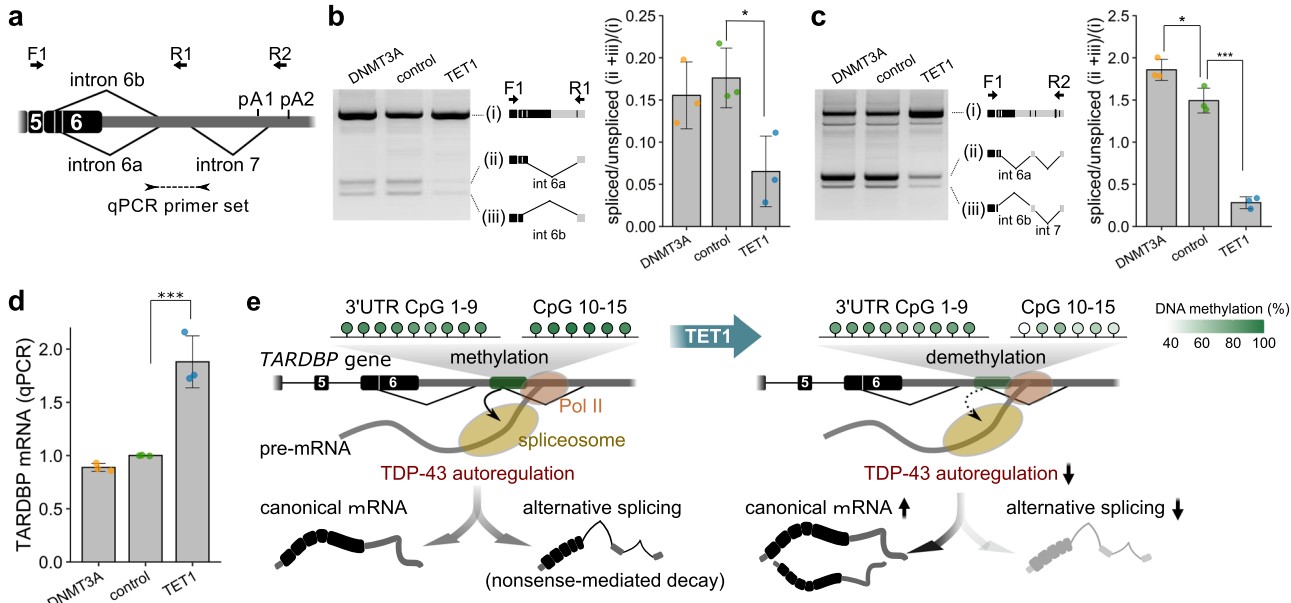

**Fig. 3 Effect of the demethylation of *TARDBP* 3′UTR CpGs on alternative splicing and mRNA expression of *TARDBP*. a** Primer sets for the RT-PCR of *TARDBP* mRNA (arrows: F1/R1, F1/F2) and quantitative real-time PCR (arrowheads). **b, c** Analysis of alternative splicing by using primer set F1/R1 (**b**) or F1/F2 (**c**) (mean ± SD, n = 3, Dunnett's test). **d** Quantitative real-time PCR analysis of *TARDBP* mRNA using *RPLP1* and *RPLP2* as reference genes (mean ± SD, n = 3, Dunnett's test). *p < 0.05, ****p < 0.001. **e** Effect of the demethylation of *TARDBP* 3′UTR CpGs 10–15 on the TDP-43 autoregulation mechanism. Green circles indicate the percent methylation of each CpG site. The binding of TDP-43 to pre-mRNA triggers alternative splicing, and this spliced isoform is degraded via nonsense-mediated mRNA decay. The amount of TDP-43 in the nucleus determines the ratio of these isoforms. However, the demethylation of 3′UTR CpGs reduces the efficiency of TDP-43 autoregulation by attenuating alternative splicing and increasing the canonical mRNA levels.

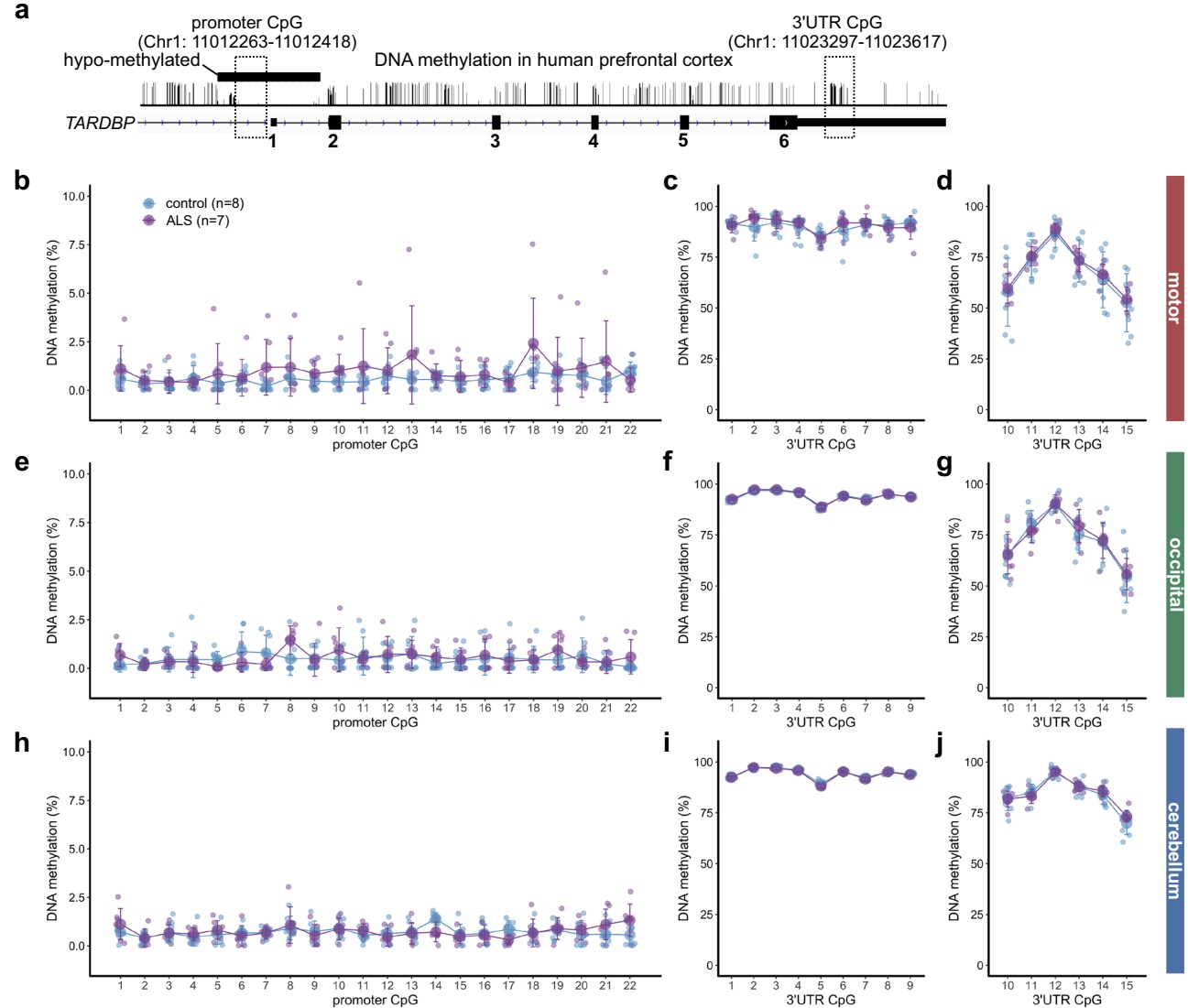

**Fig. 4 Percentages of *TARDBP* gene DNA methylation in the human brain. a** Schematic of the DNA methylation status of the human prefrontal cortex obtained from the methylomic database and the CpG region analyzed by bisulfite amplicon sequencing. **b–j** DNA methylation percentages of each CpG site in the *TARDBP* promoter region (**b**)(**e**)(**h**) and *TARDBP* 3′UTR (**c**)(**d**)(**f**)(**g**)(**i**)(**j**) in 8 controls and 7 ALS patients. Mean (large circles) ± SD.

methylated sites among 3′UTR CpGs 10–15 was extremely low (Fig. 6d). In contrast, there was no correlation with age in the other tissues or at other CpG sites (Supplementary Fig. 4). Moreover, there was no correlation with age among the ALS patients (an interaction was found in the covariance analysis, $p = 0.028$) (Fig. 6b, c). In the ALS group, the degree of DNA methylation was as low as approximately 65% at the age of 45 years and was similar to that in individuals in their 60 s and 70 s.

The epipolymorphism score of 3′UTR CpGs 10–15 was positively correlated with age in the motor cortex of the controls (Fig. 6e, Supplementary Fig. 5), which is consistent with the age-related increase in the heterogeneity of DNA methylation[36]. In addition, the methylation-linkage diagram of the two youngest (middle-aged) and two oldest (elderly) individuals showed generally lower pairwise correlations in the elderly individuals (Fig. 6f, g), suggesting stochastic demethylation of the epialleles with aging[37] rather than regulated demethylation by specific enzymes (Fig. 6h).

**Demethylation of the *TARDBP* 3′UTR increases *TARDBP* mRNA expression in the motor cortex.** Then, we investigated

the association between the DNA methylation status of the 3′UTR CpGs and *TARDBP* expression in the motor cortex. The expression level of *TARDBP* mRNA tended to increase with DNA demethylation of 3′UTR CpGs 10–15 (Fig. 7a). In particular, the analysis of only the control group without brain disease showed a significant correlation (Fig. 7a). We also examined the TDP-43 protein levels in the motor cortex of 11 controls and five ALS patients whose tissue was used in the DNA methylation analysis (Fig. 7b). In the RIPA-soluble fraction, there was no association between the DNA methylation of 3′UTR CpGs 10–15 and the TDP-43 levels (Fig. 7b, c). However, in the RIPA-insoluble urea-soluble fraction, the full-length TDP-43 protein levels (43 kDa) were inversely correlated with the percentage of DNA methylation of 3′UTR CpGs 10–15 (Fig. 7b, d). The TDP-43 C-terminal fragment (CTF) in the urea-soluble fraction also showed a similar trend in the analysis of the ALS group only (Fig. 7b, e).

**Accelerated DNA demethylation of the *TARDBP* 3′UTR is associated with the age of ALS onset.** Finally, we investigated the relationship between the DNA methylation status of the *TARDBP* 3′UTR and the clinical features of ALS patients. As shown in

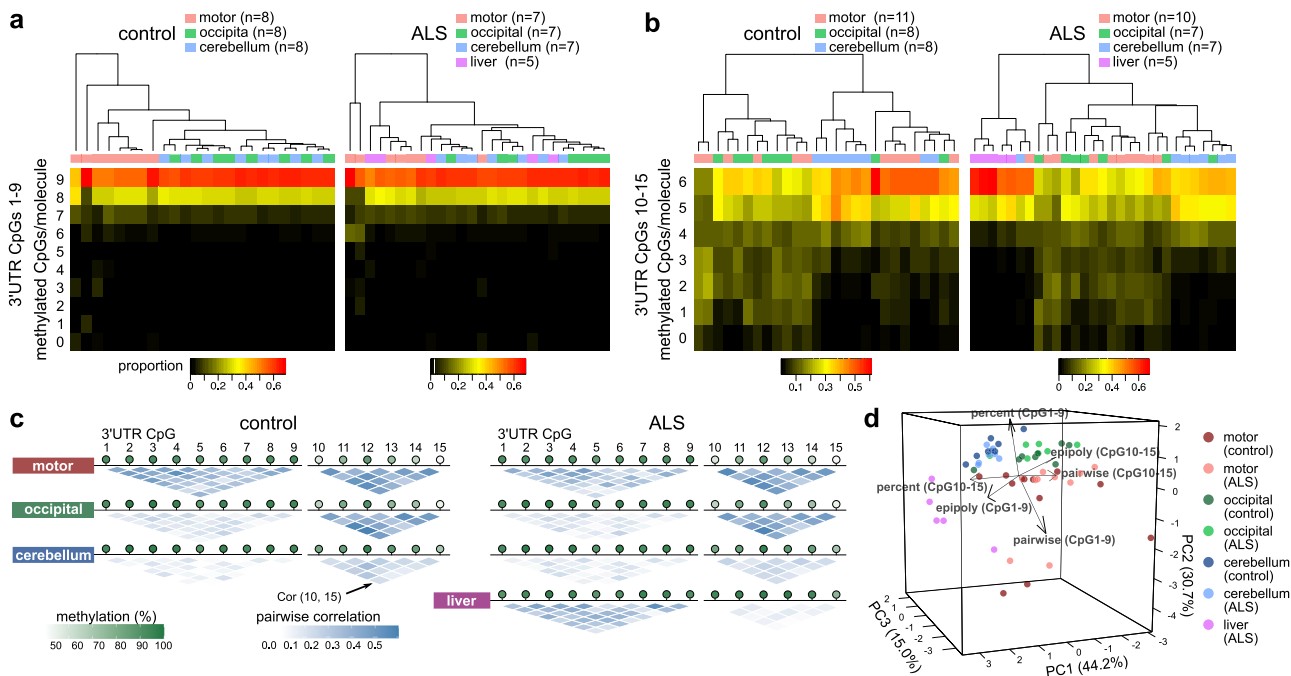

**Fig. 5 DNA methylation status of *TARDBP* in different brain regions and liver tissues. a, b** Proportions of the number of sequence reads classified by the number of CpGs methylated among nine CpG sites (3′UTR CpGs 1–9) or six CpG sites (3′UTR CpGs 10–15). **c** Methylation-linkage diagram of 3′UTR CpGs 1–9 and 3′UTR CpGs 10–15 showing the correlation of all CpG-site pairs in the target region. Green circles indicate the mean percentage of DNA methylation at each CpG site. **d** Principal component analysis using the mean methylation percentages, mean pairwise correlations, and epipolymorphism scores in 3′UTR CpGs 1–9 and 3′UTR CpGs 10–15. Eleven controls and 10 ALS patients were included in the analysis of 3′UTR CpGs 10–15 in the motor cortex, five ALS patients were included in the analysis of the liver, and seven ALS patients and eight controls were included in the other analyses.

Fig. 6, *TARDBP* 3′UTR CpGs 10–15 showed demethylation in the middle-aged ALS group compared with the control group, suggesting that the relationship between epigenetic age rather than chronological age and clinical features should be considered. Therefore, we calculated the acceleration of the DNA methylation age in the ALS cases after estimating the epigenetic age based on the DNA methylation of 3′UTR CpG 10 in the control motor cortex (Fig. 8a). As a result, the acceleration of the DNA methylation age was significantly inversely correlated with the age of ALS onset (Fig. 8b). No correlation between DNA methylation age acceleration and disease duration was observed (Fig. 8c).

## Discussion

In this study, we first showed that the 3′UTR CpGs of *TARDBP* were methylated and that the 3′UTR CpG 10–15 region was sensitive to TET1. Second, a demethylated state of 3′UTR CpGs 10–15 suppressed the alternative splicing of the *TARDBP* 3′UTR and increased the expression of *TARDBP* mRNA. Third, the 3′UTR CpG methylation pattern differed among different brain regions. Fourth, the 3′UTR was demethylated by aging in the motor cortex, and the acceleration of demethylation in ALS is related to the age of onset. Thus, we propose that age-dependent demethylation of the *TARDBP* 3′UTR in the motor cortex may partially explain why aging is a risk factor of and the motor cortex is affected in ALS.

We showed that the demethylation of 3′UTR CpGs 10–15 suppressed the splicing of intron 7. DNA methylation sites are enriched with alternative splice sites and can either enhance or suppress exon inclusion[25,26,29]. In general, DNA methylation causes RNA polymerase II pausing by modulating proteins, such as CCCTC-binding factor[23,24] and methyl-CpG-binding protein 2[26,29]. Regarding the *TARDBP* 3′UTR, excess TDP-43 halts RNA polymerase II and increases the alternative splicing of intron 7[31].

However, the CCCTC-binding factor and methyl-CpG-binding protein are not enriched in the *TARDBP* 3′UTR in public databases[30]. Therefore, we consider the possibility that proteins other than these known factors may also be involved in the mechanism linking DNA methylation and splicing.

Next, we discuss the relationship between this age-related demethylation in the motor cortex and motor cortex selectivity in ALS. We found age-dependent demethylation of 3′UTR CpGs 10–15 in the motor cortex but not the occipital and cerebellar cortexes. This result is consistent with previous studies showing that the age-related status of DNA methylation differs across brain regions[43]. In particular, our results are consistent with results showing lesser effects of age-related methylation in the occipital lobe and cerebellum[44]. The differential expression of DNMT and TET may explain the specific DNA methylation profiles associated with aging in various brain regions[45]. The expression of TET1, TET3, and DNMT3A2 is also affected by neural activity[46–48]. The accelerated demethylation of the *TARDBP* 3′UTR in the motor cortex of the ALS patients analyzed in this study was associated with the age of onset. Thus, the unique profile of *TARDBP* 3′UTR DNA methylation in the motor cortex may contribute to susceptibility to TDP-43 pathology during brain aging in individuals with ALS.

We also discuss the limitations of this study. First, our analysis targeted 15 CpGs located at the splicing site of intron 7. However, intron 6 also contributes to the autoregulation of TDP-43[13,14]. Therefore, we cannot exclude the possibility that the DNA methylation status of CpGs near the splicing site of intron 6 may be affected. Second, the present analysis included a mixture of multiple cell types. Therefore, whether the observed age dependency was due to the senescence of each cell or changes in constituent cells due to aging remains unclear[39]. Furthermore, it may be difficult to properly evaluate motor cortexes derived from ALS patients who exhibit advanced neuronal loss and gliosis in

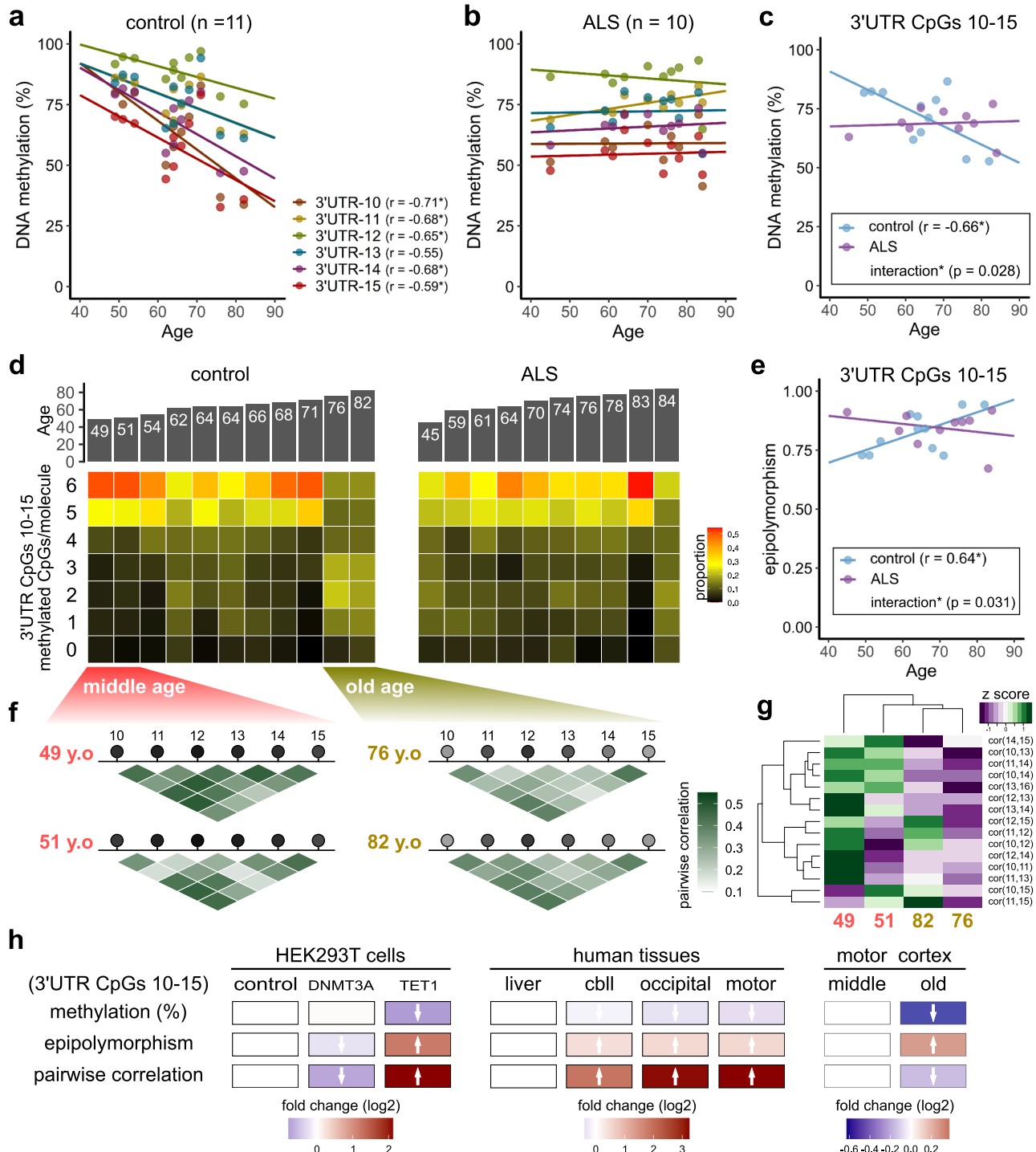

**Fig. 6 Effect of aging on the DNA methylation status of the *TARDBP* 3′UTR in the motor cortex. a, b** Scatter plot showing the percentage of DNA methylation at each CpG site among *TARDBP* 3′UTR CpGs 10–15 according to age at autopsy (Pearson's correlation test) in the control (**a**) and ALS (**b**) groups. **c** Scatterplot showing the average DNA methylation of 3′UTR CpGs 10–15 according to age at autopsy in the controls (Pearson's correlation test). **d** Proportions of the number of sequence reads classified by the number of methylated CpGs among the six CpG sites (3′UTR CpGs 10–15) in the controls and ALS patients. **e** Correlation between age at autopsy and the epipolymorphism scores of 3′UTR CpGs 10–15 (Pearson's correlation test). **f, g** Methylation-linkage diagram of 3′UTR CpGs 10–15 in two of the youngest and two of the oldest individuals analyzed (**f**) and heat map (**g**) showing the extent of the pairwise correlation of each CpG pair. **h** For each DNA methylation profile, the effects of the targeted manipulation of DNA methylation, differences according to brain region, and the effect of aging are shown as fold changes (log2) relative to the experimental control, liver, and middle-aged group results, respectively.

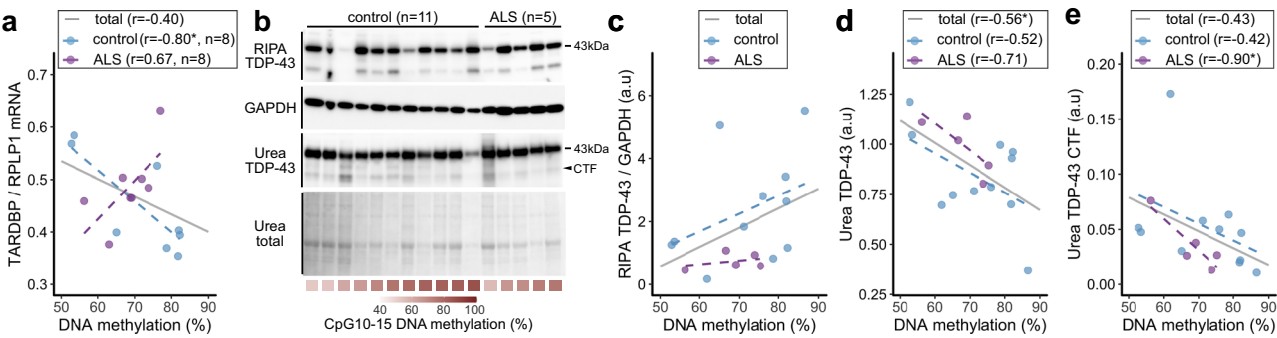

**Fig. 7 Effect of DNA methylation on *TARDBP* expression in the motor cortex. a** Scatterplot showing the association between the percentage of DNA methylation of 3′UTR CpGs 10–15 and *TARDBP* mRNA expression relative to the reference gene *RPLP1* in eight controls and seven ALS patients (a total of 15 cases). **b** Western blot analysis of the RIPA-soluble fraction and urea-soluble fraction from the motor cortex of 11 controls and five ALS patients (a total of 16 cases) using an anti-TDP-43 antibody and anti-GAPDH antibody. Total protein in the urea-soluble fraction is also shown. **c** Scatterplot showing the association between the percentage of DNA methylation of 3′UTR CpGs 10–15 and RIPA-soluble full-length TDP-43 protein (43 kDa) expression relative to the reference protein GAPDH. **d, e** Scatterplot showing the association between the percentage of DNA methylation of 3′UTR CpGs 10–15 and the urea-soluble full-length TDP-43 protein (43 kDa) (**d**) and the urea-soluble C-terminal fragment (CTF) of TDP-43 (**e**). Pearson's correlation test, *$p < 0.05$.

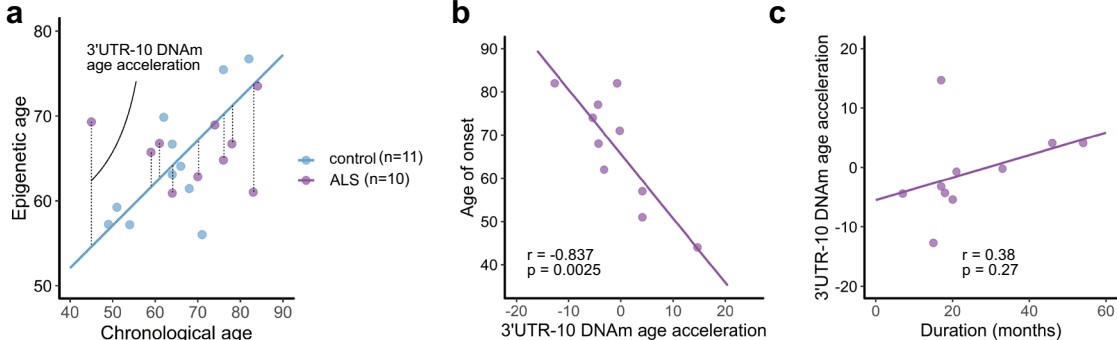

**Fig. 8 Association between DNA methylation age acceleration and the clinical features in ALS. a** "Epigenetic age" is estimated based on the regression coefficient calculated from the chronological age and the DNA methylation percentages at *TARDBP* 3′UTR CpG 10 in the control motor cortex (11 controls). The difference between the epigenetic age and the linear-regression model is defined as "DNA methylation age acceleration" in the ALS cases. Ten ALS patients were included in the analysis in relation to the clinical features. **b** Scatterplot showing the age at disease onset according to DNA methylation age acceleration at 3′UTR CpG 10. **c** Scatterplot showing the acceleration of the DNA methylation age at 3′UTR CpG 10 according to the disease duration. Pearson's correlation test.

whole tissues. In the future, single-cell methylation analyses should be performed to examine methylation changes in each cell type[22,49]. Third, the limited number of brain regions analyzed restricts our ability to emphasize tissue specificity. Finally, an analysis of a larger number of autopsy brains from patients without brain disease and ALS patients of varying ages should validate our conclusions in the future.

In summary, our findings suggest that age-related demethylation may affect gene expression levels via alternative splicing. In the central nervous system in particular, diverse mRNA isoforms produced via alternative splicing contribute to higher-order mechanisms and pathological mechanisms[50]. Therefore, the identification of the DNA methylation sites that regulate these processes and the elucidation of the underlying molecular mechanisms may contribute to understanding the pathogenesis of not only sporadic ALS but also other sporadic neurodegenerative diseases.

## Materials and methods

**Targeted manipulation of DNA methylation**. A TET1-*TARDBP*-target vector incorporating dCas9-peptide repeats and the catalytic domain of the DNA demethylase (scFv-TET1) was applied to demethylate the target DNA region (Fig. 2b)[34]. A DNMT3A-*TARDBP*-target vector incorporating dCas9 and DNMT3A, which is the catalytic domain of the DNA methyltransferase, was applied for methylation of the target DNA region (Fig. 2b)[35].

To construct the TET1-*TARDBP*-target vector, pPlatTET-gRNA2 (Addgene plasmid 82559; Addgene) was cut at the AflII site, and a guide RNA fragment was inserted using Gibson assembly master mix (New England Biolabs). Four guide RNAs were selected using Benchling (https://benchling.com/). To construct the DNMT3A-*TARDBP*-target vector, pdCas9-DNMT3A-EGFP (Addgene plasmid 71666) was cut at a BbsI site, and a guide RNA fragment was inserted in the same manner. The sequences of the guide RNAs are described in Supplementary Table 2. As a control vector, pHRdSV40-scFv-GCN4-sfGFP-VP64-GB1-NLS (Addgene plasmid 60904) was used. The plasmid vectors (0.265 pmol) were transfected using Lipofectamine 3000 (Invitrogen) into HEK293T cells cultured in Dulbecco's Modified Eagle's Medium (Gibco) containing 10% fetal bovine serum in 24-well plates. After 48 h, the GFP-positive cells were isolated using FACS Diva (BD Biosciences) software on a FACSAria II system (BD Biosciences).

**Analyzed human tissues**. Autopsied brains of sporadic ALS patients without ALS-causing gene mutations and control patients without brain disease were pathologically confirmed at the Department of Pathology, Brain Research Institute, Niigata University. The sample size was first set to eight ALS patients and eight controls following previous reports investigating the relationship between the accelerated age of DNA methylation and the age of ALS onset in the ALS autopsy brains[20,21]. During the analysis, one ALS patient was excluded due to the detection of a mutation in *TBK1*. To analyze the relationship between the DNA methylation levels of CpGs 10–15 in the motor cortex and age, we included the motor cortexes of three ALS patients and three controls and analyzed data from 10 ALS cases and 11 controls (Figs. 5–8) (Supplementary Table 1: case; ALS 8–10, control 9–11). For the five ALS patients whose frozen liver tissue was preserved, we also conducted a liver analysis (Fig. 5). Although a variant (2076 G > A) reported in the family lineage of a patient presenting with ALS and frontotemporal degeneration with elevated expression of *TARDBP*[51] caused loss of CpG 6 in the 3′UTR (Fig. 1a), we

found no variants in the patients included in this study by amplicon sequencing. The Institutional Ethical Review Board of Niigata University approved this study, which investigated postmortem tissues autopsied with written informed consent from the families.

**Analysis of the DNA methylation status.** Genomic DNA from $2.5 \times 10^5$ HEK293T cells was purified using a Nucleo Spin Tissue XS kit (MACHEREY-NAGEL). Using 25 mg of human postmortem tissue, genomic DNA was purified using a DNeasy Blood and Tissue kit (QIAGEN). According to the manufacturer's protocols, the purified genomic DNA was bisulfite treated with an EpiTect Fast DNA Bisulfite kit (QIAGEN). In both HEK293T cells and human postmortem tissues, bisulfite amplicon sequencing was performed using primer sets A and B (as shown in Fig. 2a), which target the 1st–9th (3′UTR CpGs 1–9) and the 10th–15th (3′UTR CpGs 10–15) CpG sites, respectively. The bisulfite-treated DNA was amplified by nested PCR (Supplementary Table 2) using KAPA HiFi HS Uracil + ReadyMix (Kapa Biosystems). In the first and second rounds of PCR, the thermal cycling conditions were as follows: 95 °C for 3 min, 30 cycles at 98 °C for 20 s, 60 °C for 15 s, 72 °C for 15 s, and a final step at 72 °C for 1 min. The PCR products were first diluted 1/100 with DNase-free water and then used as templates for the second round of PCR. The second-round PCR products were purified with an Agencourt AMPure XP kit (BECKMAN COULTER Life Sciences). To distinguish each sample, dual-index sequences were added to the amplified bisulfite-treated products via an additional PCR step. The index sequences were defined by the combination of primers with TruSeq HT index 1 (D 7xx) and TruSeq HT index 2 (D 5xx) (Illumina). Index PCR was performed using KAPA HiFi HS ReadyMix (Kapa Biosystems) under the following thermal cycling conditions: 95 °C for 3 min, 8 cycles at 95 °C for 30 s, 55 °C for 30 s, 72 °C for 30 s, and a final step at 72 °C for 5 min. The index PCR products were also purified with an Agencourt AMPure XP kit (BECKMAN COULTER Life Sciences).

The amplicons were sequenced on an Illumina MiSeq platform in $2 \times 251$-bp paired-end mode using MiSeq Reagent Nano Kit v2 500 cycles (Illumina). The sequenced reads were mapped to the human reference genome hg19 using the methylation analysis tool Bismark v0.18.1 (Babraham Bioinformatics)[52]. Bismark determines the methylation rates of CpG sites by dividing the total number of mapped reads by the number of reads representing the methylation status at a given CpG site. Furthermore, we generated the pileup of the sequence data at the targeted CpG sites using GATK's 'pileup' command[53]. For each fragment derived from paired-end reads, we scanned the sequence corresponding to the targeted CpG sites and counted the number of fragments in individual sequence patterns.

The epipolymorphism scores were calculated and methylation-linkage diagrams were generated based on a previous report[38]. The CpG density at each CpG site was calculated by a kernel-density estimation (bandwidth = 10.73). Principal component analysis was performed using the average methylation rates, average pairwise correlations, and epipolymorphism scores of 3′UTR CpGs 1–9 and 3′UTR CpGs 10–15. The 'pcaMethods' R package[54] was used to apply the nipals algorithm with unit-variance scaling.

**RNA analysis of HEK293T cells.** From $2.5 \times 10^5$ HEK293T cells, the total RNA was extracted using a Nucleo Spin RNA XS kit (MACHEREY-NAGEL). First-strand cDNA was synthesized from the total RNA using Prime Script Reverse Transcriptase (TAKARA Bio). RT-PCR was performed using LA Taq (TAKARA Bio). The primer pairs employed for RT-PCR are listed in Supplementary Table 2. The thermal cycling conditions were as follows: 94 °C for 2 min, 35 cycles at 94 °C for 30 s, 55 °C for 45 s, and 72 °C for 1 min and 30 s (primer pair F1–R1) or 2 min and 30 s (primer pair F1–R2), with a final step at 72 °C for 5 min. The amplified products were separated by electrophoresis on a 2% agarose gel and quantified by using Image Quant TL analysis software (GE Healthcare). Quantitative real-time PCR was performed on a TP-850 Real-Time PCR Detection System (TAKARA Bio) using SYBR Green Premix ExTaq II. The primer pairs employed for quantitative real-time PCR are listed in Supplementary Table 2. The thermal cycling conditions were as follows: 95 °C for 30 s, 40 cycles at 95 °C for 15 s and 60 °C for 30 s. The delta–delta CT method was used for the quantitative evaluation.

**RNA analysis of the human motor cortex.** One patient with an unknown postmortem interval (control 9) was excluded from the RNA analysis (Supplement Table 1). Total RNA was extracted from frozen brain tissue (100–200 mg) using a mirVana miRNA isolation kit (Applied Biosystems). The RNA quality was evaluated based on the RNA integrity number as determined using Tape Station 2200 (Agilent Technologies), and samples with RNA integrity numbers below 6.0 were also excluded. First-strand cDNA was synthesized from the total RNA using SuperScript VILO MasterMix (Invitrogen). Droplet-digital PCR was performed using QX 200 ddPCR Supermix for Probes (No dUTP) on a QX 200 Droplet Digital PCR System (Bio Rad). The primer pairs employed for PCR are listed in Supplementary Table 2. The thermal cycling conditions for the probe assays were as follows: 95 °C for 10 min, 40 cycles at 94 °C for 30 s and 55 °C for 2 min (ramping rate reduced to 2 °C per second), and a final step at 98 °C for 10 min.

**Protein extraction from the human motor cortex and western blot analysis.** The frozen motor cortex remaining after the DNA methylation analysis was homogenized in 10 volumes of ice-cold RIPA buffer (50 mM Tris-HCl, pH 7.6, 150 mM NaCl, 1% NP-40, 0.5% sodium deoxycholate, and 0.1% SDS) supplemented with a protease-inhibitor cocktail (Sigma). After sonication on ice (20%/10 sec, twice), 400 μl of lysates were centrifuged at $100,000 \times g$ at 4 °C for 30 min. The supernatant was used as a RIPA-soluble fraction. To ensure the removal of all RIPA-soluble proteins, the pellet was dissolved in RIPA buffer, sonicated (20%/10 s, twice), and centrifuged at $100,000 \times g$ for 30 min at 4 °C. The insoluble pellet was dissolved in 100 μl of urea buffer (7 M urea, 2 M thiourea, 4% CHAPS, and 30 mM Tris, pH 8.5) supplemented with a protease-inhibitor cocktail, sonicated (20%/10 s, twice), and centrifuged at $100,000 \times g$ at 22 °C for 30 min. The supernatant was used as the urea-soluble fraction. The protein concentration of the RIPA-soluble fraction was measured using a BCA protein assay kit (Thermo Fisher Scientific). The protein concentration of the urea-soluble fraction was assumed to have the same ratio as the RIPA-soluble fraction. The supernatant was resuspended in Laemmli Sample Buffer (BioRad) and treated at 96 °C for 5 min. The proteins were separated by SDS-PAGE using a 10% polyacrylamide gel (Super SepTM Ace, Wako) and transferred to a PVDF membrane (Millipore). After transfer, total protein on the membrane was visualized using No-Stain Protein Labeling Reagent (Thermo Fisher Scientific). The blot was immersed in the primary antibody overnight at 4 °C. The antibodies used for immunoblotting included a rabbit anti-TDP-43 (C-terminal) polyclonal antibody (Proteintech 12892-1-AP, 1:5000 for the RIPA-soluble fraction, 1:2000 for the urea-soluble fraction) and a mouse anti-GAPDH monoclonal antibody (MBL). HRP-conjugated secondary antibodies (Dako) were used. The bands were detected using Immobilon Western Chemiluminescent HRP Substrate (Millipore) and quantitatively analyzed using Amersham Imager 680 (GE Healthcare).

**Statistics and reproducibility.** For each experiment, the corresponding statistical test, which was two-tailed, is indicated in each figure legend. The number of samples per group is indicated in each figure legend. Statistical significance was considered to exist at $p < 0.05$.

**Reporting summary.** Further information on research design is available in the Nature Research Reporting Summary linked to this article.

## Data availability
Source data are provided with this paper as "Supplementary Data 1". All relevant data are available from the corresponding author upon reasonable request.

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

## Acknowledgements

This research was supported by a grant-in-aid for Scientific Research on Innovative Areas (Brain Protein Aging and Dementia Control; 26117006) from MEXT, grants-in-aid for Scientific Research (A) (26250017 and 25253065), Scientific Research (C) (17K09751), and Young Scientists (19K23961) from JSPS; a grant-in-aid from the Research Committee of CNS Degenerative Diseases and Comprehensive Research on Disability, Health, and Welfare (13230021) from the Japanese Ministry of Health, Labour, and Welfare of Japan; the Tsubaki Memorial Foundation; the Takeda Science Foundation; and SERIKA FUND.

## Author contributions

YK, AS, and OO designed this study and wrote the paper. YK and AS conducted the experiments. YK and AS analyzed the results. NH and TI contributed to the experiments and analyses using the MiSeq platform (Illumina). JI, MT, and AK contributed to the provision of human-autopsied tissues. AY, TI, TY, and ST contributed to the interpretation of the results. All authors critically revised the draft and approved the final version.

## Competing interests

The authors declare no competing interests.
