## [Peer Review File · Communications Biology]

Reviewers' comments:

Reviewer #1 (Remarks to the Author):

An important subject: TDP-43 expression level. Good introduction, excellent hypothesis on the basis of published data linking TDP-43 gene methylation and age-related neurodegenerative disease. Specifically, autoregulation. Epigenetic effect on autoregulation is striking and very important and a very novel take, with some a nice dCas9 approach. This give a very robust readout.

Some comments:

1. A general comment: they ought to cite work that directly links failure of autoregulation to disease mutations e.g. White et al Nat Nsci 2018
2. page 2, bottom – I appreciate what they are saying but I am not sure they can use the word 'excess'. We do not know that it is an excess of tdp-43 that triggers autoregulation. Autoregulation is likely to be at play all the time to a greater or lesser extent.
3. Aging study is very interesting. Have they considered looking at other genes as well to see if the methylation differences between controls and ALS are just related to TDP-43?
4. Fig 6G spelling mistake "midle age"
5. I don't understand why in fig 6j they haven't got data from all their controls and ALS cases.
6. What is their explanation for the difference in epigenetic modification effects on TARDBP expression between controls and ALS? (6j) They provide only minor discussion.
7. Have they considered that there must be some benefit to age-related demethylation of the UTR in the motor cortex... If so, what do they think this is??
8. can they speculate as to how their findings should direct therapeutic development? Are they able to use their tools in a screen?

Reviewer #2 (Remarks to the Author):

In this work, the authors described how DNA demethylation in the autoregulatory region of TDP-43 reduced alternative splicing and increased TDP-43 expression in the human motor cortex, we found that this region was demethylated with age and that the expression of TDP-43 increased. The dysregulation of TDP-43 autoregulation by age-related DNA demethylation in the motor cortex may explain the contribution of aging and system selectivity in ALS. They used a nice system to verify if the methylation status determines changes in alternative splicing.

Few minor revisions:

- In the abstract, aging is repeated too many times;
- The authors are stating : " we found that this region was demethylated with age". Could you please justify better this, since you used only 7 samples (figure 6a-c) and 10 in figure 6f?
- Can you justify the number of the samples used with a power analysis?

Reviewer #3 (Remarks to the Author):

It has been shown that DNA methylation influences alternative splicing. The manuscript aimed to determine (1) whether the methylation status of TARDBP, a causal gene for ALS, contributes to the usage of an alternative spliced exon within the 3'-UTR, and (2) whether 3'-UTR methylation of

TARDBP is altered in the motor cortex during aging and in ALS. The spliced exon within the 3'-UTR of TARDBP produce mRNAs that are subjected to nonsense-mediated decay. This has been shown to be one of the autoregulatory mechanisms to regulate TDP-43 (encoded by TARDBP) expression. Using publicly available data, the authors found that many CpGs, which play a prominent role in DNA-methylation, are clustered around the alternative splicing site in the TARDBP 3'-UTR and that these CpG sites are moderately to highly methylated in the human prefrontal cortex. This observation led them to hypothesize that the DNA methylation status influence the alternative splicing within the 3'-UTR of TARDBP, thereby affecting TDP-43 expression level. Using the dCas9 system, the authors manipulated the DNA methylation status by targeting either Tet Methylcytosine Dioxygenase 1 (TET1), which converts methylcytosine to hydroxymethylcytosine, an intermediate for unmethylated cytosine, and DNMT3A, a methyltransferase, to the 3'-UTR of TARDBP gene in HEK293 cells. They found that there are two distinct groups of CpGs surrounding the known TDP-43 binding sites: group 1 (CpG 1-9) and group 2 (CpGs 10-15). Intriguingly, only group 2 (CpGs 10-15) are subjected to TET1-mediated demethylation. Functionally, demethylation of CpGs 10-15 led to less spliced events for intron 6 and intron 7 of TARDBP 3'-UTR CpGs. The authors suggest that demethylation of CpGs 10-15 could be one of the mechanisms contributing to increased TDP-43 level, as demethylation of CpGs 10-15 favours the productive generation of TARDBP mRNA. actually suppressed the splicing of intron 6 and intron 7 of TARDBP and increased TARDBP mRNA levels. Transferring their investigations to the human brain, the authors investigated DNA methylation levels of TARDBP in the motor, occipital, and cerebellar cortexes of control and sporadic ALS patients. The methylation pattern at the 3'UTR CpG of TARDBP differed among the brain regions, with human motor cortex demethylation increasing with age. Although it does not appear that the methylation status of CpGs 10-15 corelates with ALS, the authors claims that there is an age-dependent decrease of CpGs 10-15 methylation in the motor cortex of "control" subjects. As aging is considered a risk factor for ALS, the authors concluded that the age-dependent demethylation of the TARDBP 3'-UTR in the human motor cortex may explain why the motor cortex is selectively affected in ALS patients.

Overall, the manuscript is well-written and a pleasure to read. The cell experiments were done well with proper controls. This is an interesting and potentially important study. It is unfortunate that the results from ALS patients do not quite support the attractive model that the author proposed. The manuscript could be of interest for readers of Communication Biology. This referee have a few comments and suggestions for the authors to consider:

Major comments and suggestions:

1. There have been at least two models to explain how DNA methylation may influence alternative splicing: the kinetic model and the recruit model (Luco et al., Cell, 2011; DOI: 10.1016/j.cell.2010.11.056; Lev Maor et al., Trends in Genetics, doi: 10.1016/j.tig.2015.03.002). What the authors proposed are consistent with the kinetic model and make intuitive sense (at least to this referee), where DNA methylation slows down RNA polymerase II and allows the splicing to occurs. However, the genomic study by Ast group suggests that the methylation status could both positively (enchancing) or negatively (inhibiting) regulate splicing (Yearim et al., Cell Report, 2015 , doi: 10.1016/j.celrep.2015.01.038). In addition, based on the literature, it is likely that the splicing can be mediated by CCCTC-binding factor (CTCF) (Shukla et al., Nature, 2011, doi: 10.1038/nature10442), MeCP2 (Maunakea et al., Cell Research, 2013, doi: 10.1038/cr.2013.110), or HP1 (Yearim et al., Cell Report, 2015). Is it possible for the authors to perform ChIP experiments with CTCF, MeCP2 and HP1 and check the binding to TARDBP gene (by qPCR after ChIP) in the TET1 transfected cells? This could provide some potential mechanistic view. In addition, could the authors perform immunoblots for TDP-43 to determine whether TDP-43 protein is increased in the TET1- transfected cells? This will strengthen the authors' argument.
2. While it is interesting that there is an apparent age-dependent methylation decrease in the human motor cortex, this referee would like to caution that the "n" is relative small with n=1 per age point. Is it possible to have more data points? If it is not possible, I would suggest that the authors to use caution on the conclusion, i.e., a large scale of normal aging group should be used to confirm this hypothesis/model, etc. In addition, is it possible to include the liver data from control subjects? It is a good control/contrast group, which was included for ALS patients. Furthermore, is TDP-43 level increased in the aged brains? The authors should perform qRT-PCR/immunoblots for TDP-43 for these samples to establish the correlation between demethylation

and TDP-43 level.

3. For the analysis of ALS patients, is it possible to include the data from other methylation study, e.g., Zhang et al., *Acta Neuropathologica*, 2020 (doi: 10.1007/s00401-020-02131-z)? Similar to the previous points, could qRT-PCR/immunoblots for TDP-43 for these ALS samples be performed?

Other minor comments:

1. Please elaborate on the gRNA(-) TET1 construct. Is this with "negative" gRNA or no gRNA, etc?
2. Could the author elaborate/rephrase line 88-90? It is not obvious what the authors are trying to say.
3. Although the numbers of human samples in Figure 5 were stated in the figure legend, would it be possible to indicate the numbers in each panel of Figure 5? The referee's intention is to clearly display the numbers of human samples in each analysis.
4. There are typographical errors in Materials and Methods section:
 - a. Line 210: "Dulbecco's Modified Eagle's Medium" should all start with a capital letter.
 - b. Line 218: "causes the loss of the sixth of the 15 CpG sites in the 3'UTR" is this supposed to be "causes the loss of one-sixth of the 15 CpG sites in the 3'UTR"
 - c. Line 235 and 242: "AMpure" should be AMPure
 - d. Line 238-239: "D 7XX" and "D 5XX" should be "D7xx" and "D5xx" respectively
 - e. Figure 6e: "midle age" should be "middle age"
 - f. Supplementary Table 2:
 - i. "targetting" should be "targeting"
 - ii. "drolet" should be "droplet"

Reviewer #1 (Remarks to the Author):

An important subject: TDP-43 expression level. Good introduction, excellent hypothesis on the basis of published data linking TDP-43 gene methylation and age-related neurodegenerative disease. Specifically, autoregulation. Epigenetic effect on autoregulation is striking and very important and a very novel take, with some a nice dCas9 approach. This give a very robust readout.

Some comments:

Comment 1:

A general comment: they ought to cite work that directly links failure of autoregulation to disease mutations e.g. White et al Nat Nsci 2018.

Response 1:

We appreciate these meaningful suggestions by the reviewer.

According to this suggestion, we cited two studies (White, et al. Nat Neurosci 2018, Fratta, et al. EMBO 2018) and added the following sentence: "Some mutations in the TARDBP gene can alter alternative splicing and affect the autoregulatory mechanism."

Comment 2:

page 2, bottom – I appreciate what they are saying but I am not sure they can use the word 'excess'. We do not know that it is an excess of tdp-43 that triggers autoregulation. Autoregulation is likely to be at play all the time to a greater or lesser extent.

Response 2:

We agree that the word 'excess' is not appropriate in this context and revised the manuscript as follows "Nuclear TDP-43 binds the TARDBP pre-mRNA 3'UTR, and the level of TDP-43 is strictly autoregulated via its alternative splicing¹¹⁻¹³. An increased level of nuclear TDP-43 promotes its splicing to produce nonsense-mediated mRNA decay-sensitive TARDBP mRNA and reduce the level of TDP-43^{12,14}. However, when the level of TDP-43 in the nucleus is reduced, this splicing is repressed, and the TARDBP mRNA levels are increased."

Comment 3:

Ageing study is very interesting. Have they considered looking at other genes as well to see if the methylation differences between controls and ALS are just related to TDP-43?

Response 3:

We appreciate this important comment. It has been reported that DNA methylation age acceleration is associated with the age of onset in ALS patients with *C9orf72* extension (Zhang, Acta Neuropathol 2017). Combined with a report showing a relationship between DNA methylation age acceleration and the age of onset in sporadic ALS (Zhang, Acta Neuropathol 2020), this work provides a reference for us to deeply analyze the effect of aging. Therefore, we rewrote the sentence in the fourth paragraph in the *Introduction* as follows: "Indeed, an accelerated epigenetic age based on DNA methylation is associated with the age of onset of ALS patients with *C9orf72* expansion repeats²⁰ and sporadic ALS patients²¹."

Comment 4:

Fig 6G spelling mistake "midle age"

Response 4:

We corrected the spelling mistake and labeled it 'middle age' (the corresponding figure was changed from Fig. 6G to Fig. 6d).

Comment 5:

I don't understand why in fig 6j they haven't got data from all their controls and ALS cases.

Response 5:

For an accurate RNA analysis, we excluded one case with an unknown postmortem interval (control9) and five cases with RNA integrity numbers below 6.0 (ALS6, ALS8, control11, control8, and control9) among the cases used for the DNA methylation analyses (Supplementary Table 1). We mentioned this issue in the 'RNA analysis of the human motor cortex' section (*Materials and methods*) as follows: "One patient with an unknown postmortem interval (control 9) was excluded from the RNA analysis (Supplement Table 1)" and "The RNA quality was evaluated based on the RNA integrity number as determined using Tape Station 2200 (Agilent Technologies, Santa Clara, CA, USA), and samples with RNA integrity numbers below 6.0 were also excluded."

Comment 6:

What is their explanation for the difference in epigenetic modification effects on TARDBP expression between controls and ALS? (6j) They provide only minor discussion.

Response 6:

We agree that further discussion regarding this issue is warranted. In ALS-affected tissues, the expression and localization of several RNA-binding proteins that may affect RNA metabolism are disturbed. In addition, neuronal loss or gliosis may affect the expression levels of *TARDBP* mRNA measured in tissues. Therefore, the effect of the DNA methylation of the *TARDBP* 3'UTR may be ambiguous in ALS. However, in the present study, there was not enough evidence to definitively support these effects. Furthermore, it is possible that the power of detection was insufficient due to the small number of cases. Therefore, we rewrote the fifth paragraph of the *discussion* as follows: “Furthermore, it may be difficult to properly evaluate the motor cortex derived from ALS patients, who exhibit advanced neuronal loss and gliosis with whole tissues. In the future, methylation analyses at the single-cell level should be performed to examine the methylation changes in each cell type” and “Finally, an analysis of a larger number of autopsy brains from patients without brain disease and ALS patients of varying ages should validate our conclusions in the future.”

Comment 7:

Have they considered that there must be some benefit to age-related demethylation of the UTR in the motor cortex... If so, what do they think this is??

Response 7:

We appreciate this question from a new perspective. Indeed, based on the prerevision results alone, it was not possible to determine whether the demethylation of the *TARDBP* 3'UTR is an age-related disruption or a beneficial response to aging. However, our further analysis revealed that the more demethylated the *TARDBP* 3'UTR was in the motor cortex, the more insoluble TDP-43 was increased. Furthermore, an acceleration of demethylation is associated with an earlier onset of disease in ALS patients. We added the results of these analyses in Fig. 7 and Fig. 8. Thus, the demethylation of the 3'UTR with aging suggests a disruptive event rather than any advantage.

Fig.7

Fig.8

Comment 8:

Can they speculate as to how their findings should direct therapeutic development? Are they able to use their tools in a screen?

Response 8:

The manipulation of DNA methylation using dCas9 to target the DNA methylation state of the *TARDBP* 3'UTR could be a potential therapeutic strategy. However, many issues still need to be resolved. Additionally, if age-related changes in the DNA methylation of the *TARDBP* 3'UTR can be detected in easily accessible cells, such as blood cells, it could serve as a biomarker of an accelerated DNA methylation age associated with ALS. We would like to consider this possibility as an avenue for our future research. Since this discussion extends beyond the findings of this study, however, we would prefer to refrain from adding this discussion to the main text.

Reviewer #2 (Remarks to the Author):

In this work, the authors described how DNA demethylation in the autoregulatory region of TDP-43 reduced alternative splicing and increased TDP-43 expression in the human motor cortex, we found that this region was demethylated with age and that the expression of TDP-43 increased. The dysregulation of TDP-43 autoregulation by age-related DNA demethylation in the motor cortex may explain the contribution of aging and system selectivity in ALS. They used a nice system to verify if the methylation status determines changes in alternative splicing.

Few minor revisions:

Comment 1:

-In the abstract, aging is repeated too many times;

Response 1:

We appreciate this advice regarding improving our paper. We carefully rewrote the abstract while avoiding repeating the word "aging".

Comment 2:

-The authors are stating:" we found that this region was demethylated with age". Could you please justify better this, since you used only 7 samples (figure 6a-c) and 10 in figure 6f?

Response 2:

We performed the power analysis as follows. We referred to previous reports investigating the association between DNA methylation age acceleration and the age of onset in ALS patients with *C9orf72* repeat expansion (correlation coefficient: -0.935) (Zhang, Acta Neuropathol 2017) and sporadic ALS patients (correlation coefficient: -0.806) (Zhang, Acta Neuropathol 2020) both in the spinal cord, which is one of the affected regions in ALS. Based on these effect sizes in the correlation analyses, we estimated an effect size of 0.8-0.9 in the motor cortex and calculated the sample size by setting the power to 0.8 and the significance level to 0.05. As a result, the required sample sizes were calculated to be 7-9. In addition, since one of the initially recruited ALS cases was found to have a *TBK1* mutation during the analyses, we excluded this case. Even in the analysis of the 8 control cases, there was an inverse correlation between age and the DNA methylation percentages (right figure). Nevertheless, because of the high interindividual variability of the DNA methylation levels in the motor cortex, we added 3 ALS cases and 3 control cases to the DNA methylation analyses of the

TARDBP 3'UTR CpGs 10-15 for a more reliable assessment. We rewrote the 'Analyzed human tissues' section (*Materials and methods*) and added the above case selection process as follows:

“The sample size was first set to 8 ALS patients and 8 controls following previous reports investigating the relationship between the accelerated age of DNA methylation and the age of onset in ALS autopsy brains (Zhang, Acta Neuropathol 2017, Zhang, Acta Neuropathol 2020). During the analysis, one ALS patient was excluded due to the detection of a mutation in *TBK1*. To analyze the relationship between the DNA methylation levels of CpGs 10-15 in the motor cortex and age, we included the motor cortex of 3 ALS patients and 3 controls and analyzed the

data of 10 ALS cases and 11 controls (Figs. 5-8) (Supplementary Table 1: Case; ALS 8–10, control 9–11)”. The number of samples analyzed was also added to the figures. We also considered adding more cases for this revision, but unfortunately, we were unable to obtain an appropriate series of control cases for the DNA methylation analyses because of the limited accumulation of frozen human autopsy brains without brain diseases.

Comment 3:

- Can you justify the number of the samples used with a power analysis?

Response 3:

Please refer to Response 2.

Reviewer #3 (Remarks to the Author):

It has been shown that DNA methylation influences alternative splicing. The manuscript aimed to determine (1) whether the methylation status of TARDBP, a causal gene for ALS, contributes to the usage of an alternative spliced exon within the 3'-UTR, and (2) whether 3'-UTR methylation of TARDBP is altered in the motor cortex during aging and in ALS. The spliced exon within the 3'-UTR of TARDBP produce mRNAs that are subjected to nonsense-mediated decay. This has been shown to be one of the autoregulatory mechanisms to regulate TDP-43 (encoded by TARDBP) expression. Using publicly available data, the authors found that many CpGs, which play a prominent role in DNA-methylation, are clustered around the alternative splicing site in the TARDBP 3'-UTR and that these CpG sites are moderately to highly methylated in the human prefrontal cortex. This observation led them to hypothesize that the DNA methylation status influence the alternative splicing within the 3'-UTR of TARDBP, thereby affecting TDP-43 expression level. Using the dCas9 system, the authors manipulated the DNA methylation status by targeting either Tet Methylcytosine Dioxygenase 1 (TET1), which converts methylcytosine to hydroxymethylcytosine, an intermediate for unmethylated cytosine, and DNMT3A, a methyltransferase, to the 3'-UTR of TARDBP gene in HEK293 cells. They found that there are two distinct groups of CpGs surrounding the known TDP-43 binding sites: group 1 (CpG 1-9) and group 2 (CpGs 10-15). Intriguingly, only group 2 (CpGs 10-15) are subjected to TET1-mediated demethylation. Functionally, demethylation of CpGs 10-15 led to less spliced events for intron 6 and intron 7 of TARDBP 3'-UTR CpGs. The authors suggest that demethylation of CpGs 10-15 could be one of the mechanisms contributing to increased TDP-43 level, as demethylation of CpGs 10-15 favours the productive generation of TARDBP mRNA.

actually suppressed the splicing of intron 6 and intron 7 of TARDBP and increased TARDBP mRNA levels. Transferring their investigations to the human brain, the authors investigated DNA methylation levels of TARDBP in the motor, occipital, and cerebellar cortexes of control and sporadic ALS patients. The methylation pattern at the 3'UTR CpG of TARDBP differed among the brain regions, with human motor cortex demethylation increasing with age. Although it does not appear that the methylation status of CpGs 10-15 correlates with ALS, the authors claims that there is an age-dependent decrease of CpGs 10-15 methylation in the motor cortex of "control" subjects. As aging is considered a risk factor for ALS, the authors concluded that the age-dependent demethylation of the TARDBP 3'-UTR in the human motor cortex may explain why the motor cortex is selectively affected in ALS patients.

Overall, the manuscript is well-written and a pleasure to read. The cell experiments were done well with proper controls. This is an interesting and potentially important study. It is unfortunate that the results from ALS patients do not quite support the attractive model that the author proposed. The manuscript could be of interest for readers of Communication Biology. This referee have a few comments and suggestions for the authors to consider:

Major comments and suggestions:

Comment 1-1:

There have been at least two models to explain how DNA methylation may influence alternative splicing: the kinetic model and the recruit model (Luco et al., Cell, 2011; DOI: 10.1016/j.cell.2010.11.056; Lev Maor et al., Trends in Genetics, doi: 10.1016/j.tig.2015.03.002). What the authors proposed are consistent with the kinetic model and make intuitive sense (at least to this referee), where DNA methylation slows down RNA polymerase II and allows the splicing to occurs. However, the genomic study by Ast group suggests that the methylation status could both positively (enchancing) or negatively (inhibiting) regulate splicing (Yearim et al., Cell Report, 2015 , doi: 10.1016/j.celrep.2015.01.038). In addition, based on the literature, it is likely that the splicing can be mediated by CCCTC-binding factor (CTCF) (Shukla et al., Nature, 2011, doi: 10.1038/nature10442), MeCP2 (Maunakea et al., Cell Research, 2013, doi: 10.1038/cr.2013.110), or HP1 (Yearim et al., Cell Report, 2015).

Is it possible for the authors to perform ChIP experiments with CTCF, MeCP2 and HP1 and check the binding to TARDBP gene (by qPCR after ChIP) in the TET1 transfected cells? This could provide some potential mechanistic view.

Response 1-1:

We appreciate these meaningful suggestions by the reviewer.

As the reviewer noted, it is important to clarify the mechanism linking DNA methylation and splicing of the *TARDBP* 3'UTR by a ChIP analysis. Prior to the ChIP experiment, we first checked the affinities of CCCTC-binding factor (CTCF), methyl-CpG binding protein (MeCP2) and heterochromatin protein 1 (HP1) to the *TARDBP* 3'UTR in a public database (ChIP-Atlas; Oki et al, EMBO Rep 2018). Referring to the database, none of these factors were enriched in the *TARDBP* 3'UTR (see below). We added this information in the second paragraph of the *Discussion* as follows: “In general, DNA methylation causes RNA polymerase II to pause by modulating proteins, such as CCCTC-binding factor^{20,21} and methyl-CpG binding protein^{23,26}. Regarding the *TARDBP* 3'UTR, excess TDP-43 halts RNA polymerase II and increases the alternative splicing of intron 7²⁸. However, CCCTC-binding factor and methyl-CpG binding protein are not enriched in the *TARDBP* 3'UTR in public databases²⁷. Therefore, we consider the possibility that proteins other than these known factors may also be involved in the mechanism linking DNA methylation and splicing.”

Therefore, we conducted ChIP experiments using HEK293T cells with TET1-mediated demethylation focusing only on RNA polymerase II and H3K27ac, which were enriched in the *TARDBP* 3'UTR in public databases. We used HEK293T cells transiently expressing TET1 without GFP sorting to obtain sufficient cell numbers for the ChIP analysis. Unfortunately, this assay did not yield reliable data for either RNA polymerase II or H3K27ac upon TET1-mediated demethylation. Therefore, we refrained from adding these data in this revision due to the limitations of our current experimental system. We assume that multiple molecules and histone modifications are involved in the mechanism linking DNA methylation and alternative splicing. Therefore, we recognize that several steps in carefully controlled experiments are necessary to elucidate the mechanism. We will attempt to clarify this issue in the future.

However, based on this important suggestion, we revised the schematic diagram (Fig. 3e). Since the previous diagram might mislead the reader and suggest that the binding affinity of the TDP-43 protein to *TARDBP* RNA does not change with DNA demethylation, we revised the text to indicate that we have no experimental data regarding the binding affinity of TDP-43. We also aim to reveal the contribution of DNA demethylation to TDP-43's binding affinity to *TARDBP*

RNA in the future.

Comment 1-2:

In addition, could the authors perform immunoblots for TDP-43 to determine whether TDP-43 protein is increased in the TET1- transfected cells? This will strengthen the authors' argument.

Response 1-2:

We are very grateful for this important suggestion. We would first like to mention the results of the western blot analysis of the motor cortex that remained after the DNA methylation analysis (the details are described in Response 2-3 below). We found that the amount of TDP-43 in the RIPA-insoluble urea-soluble fraction, but not the RIPA-soluble fraction, was increased by DNA demethylation of the *TARDBP* 3'UTR (Fig. 7 b-e).

Fig.7

This finding was consistent with the previously reported results of TDP-43 protein expression when endogenous *TARDBP* mRNA expression was increased in the mouse spinal cord (Sugai, *et al.* Neurobiology of Disease 2019). Therefore, we thought that the changes in the RIPA-insoluble urea soluble fraction could be important for confirming the alteration in the TDP-43 amounts by TET1-mediated demethylation in HEK293T cells. We transfected HEK293T cells with the TET1-*TARDBP*-target vector, DNMT-*TARDBP*-target vector, and control vector, and after 48 hours, the GFP-positive cells were isolated using FACS for the

protein extraction and western blot analysis. However, TDP-43 expression was undetectable in the RIPA-insoluble urea-soluble fraction. In addition, there were no differences in the amount of TDP-43 in the RIPA soluble fraction between the TET1-*TARDBP*-target vector- and control vector-transfected HEK293T cells (figure below).

Similar to the ChIP experiments, we consider the limitation of the analyses using cultured cells with the transient expression of TET1. Based on the results in the human motor cortex, we are uncertain whether the data in the RIPA-soluble fraction alone can be used to represent the change in the amount of TDP-43 by TET1-mediated demethylation. Therefore, we prefer to refrain from adding these western blot data to the revised manuscript.

Comment 2-1:

While it is interesting that there is an apparent age-dependent methylation decrease in the human motor cortex, this referee would like to caution that the “n” is relative small with n=1 per age point. Is it possible to have more data points? If it is not possible, I would suggest that the authors to use caution on the conclusion, i.e., a large scale of normal aging group should be used to confirm this hypothesis/model, etc.

Response 2-1:

We appreciate the recommendation for improving the quality of this paper. We considered adding more cases in this revision, but unfortunately, we were unable to obtain an appropriate series of control cases for the DNA methylation analyses because of the limited accumulation of frozen human autopsy brains without brain diseases. The power analysis that led to our sample size determination is shown below. We referred to previous reports investigating the association between DNA methylation age acceleration and the age of onset in ALS patients with *C9orf72* repeat expansion (correlation coefficient: -0.935) (Zhang, Acta Neuropathol 2017) and sporadic ALS patients (correlation coefficient: -0.806) (Zhang, Acta Neuropathol 2020) both in the spinal cord, which is one of the affected regions in ALS. Based on these effect sizes in the correlation analyses, we estimated an effect size of 0.8-0.9 in the motor cortex and calculated the sample size by setting the power to 0.8 and the significance level to 0.05. As a result, the required

sample size of ALS cases was calculated to be 7-9. In addition, since one of the initially recruited ALS cases was found to have a *TBK1* mutation during the analyses, we excluded this case. Even in the analysis using 8 control cases and 7 ALS cases, there was an inverse correlation between age and the DNA methylation percentages in the control cases (right figure).

To further provide a more reliable assessment of the correlation between the DNA methylation status of the motor cortex and age, we added the motor cortices of three ALS cases and three control cases to the analysis in the prerevision manuscript. We rewrote the 'Analyzed human tissues' section (*Materials and methods*) and added the above case selection process as follows: “The sample size was first set to 8 ALS patients and 8 controls following previous reports investigating the relationship between the accelerated age of DNA methylation and the age of onset in ALS autopsy brains (Zhang, *Acta Neuropathol* 2017, Zhang, *Acta Neuropathol* 2020). During the analysis, one ALS patient was excluded due to the detection of a mutation in *TBKI*. To analyze the relationship between the DNA methylation levels of CpGs 10-15 in the motor cortex and age, we included the motor cortex of 3 ALS patients and 3 controls and analyzed the data of 10 ALS cases and 11 controls (Figs. 5-8) (Supplementary Table 1: Case; ALS 8–10, control 9–11)”.

Nonetheless, we agree that an analysis with a larger sample size is important, including a correlation analysis with *TARDBP* expression. Therefore, we added the following limitation of this study to the Discussion: “ Finally, an analysis of a larger number of autopsy brains from patients without brain disease and ALS patients of varying ages should validate our conclusions in the future.”

Comment 2-2:

In addition, is it possible to include the liver data from control subjects? It is a good control/contrast group, which was included for ALS patients.

Response 2-2:

Unfortunately, we cannot analyze the liver tissues of the control cases used for the DNA methylation analyses because these tissues are not available. To clarify this issue, we added the following sentence to the *Materials and methods*: “Regarding the five ALS patients whose frozen liver tissue was preserved, we also conducted a liver analysis (Fig. 5).”

Comment 2-3:

Furthermore, is TDP-43 level increased in the aged brains? The authors should perform qRT-PCR/immunoblots for TDP-43 for these samples to establish the correlation between demethylation and TDP-43 level.

Response 2-3:

We agree with this proposal. First, we already show an inverse correlation between DNA methylation and the *TARDBP* mRNA levels in the motor cortex of control patients without brain disease. A similar relationship was not evident in the ALS cases. “Then, we investigated the association between the DNA methylation status of the 3'UTR CpGs and *TARDBP* expression in the motor cortex. The expression level of *TARDBP* mRNA tended to increase with DNA demethylation of 3'UTR CpGs 10–15 (Fig. 7a). In particular, the analysis of only the control group without brain disease showed a significant correlation (Fig. 7a).” There are several possible reasons why the results were not as hypothesized in the ALS motor cortex. In ALS-affected tissues, the expression and localization of several RNA-binding proteins that may affect RNA metabolism are disturbed. In addition, neuronal loss or gliosis may affect the expression levels of *TARDBP* mRNA measured in tissues. Therefore, the effect of the DNA methylation of the *TARDBP* 3'UTR may be ambiguous in ALS. However, in the present study, there was not enough evidence to definitively state these effects. Rather, it is possible that the power of detection was insufficient due to the small number of cases. Therefore, we rewrote the fifth paragraph of the *discussion* as follows: “it may be difficult to properly evaluate the motor cortex derived from ALS patients, who exhibit advanced neuronal loss and gliosis with whole tissues. In the future, methylation analyses at the single-cell level should be performed to examine the methylation changes in each cell type” and “Finally, an analysis of a larger number of autopsy brains from patients without brain disease and ALS patients of varying ages should validate our conclusions in the future.”

In this revision, we performed a western blot analysis of tissue blocks (11 controls and 5 ALS patients) that were retained after being used for the DNA methylation analysis. As a result, we found that the demethylation of *TARDBP* 3'UTR CpGs 10-15 increased the amount of TDP-43 (full-length TDP-43) in the RIPA-insoluble urea-soluble fractions in the human motor cortex, although there was no correlation with the amount of TDP-43 in the RIPA-soluble fractions. Moreover, we showed that the C-terminal fragment (CTF) of TDP-43 in the RIPA-insoluble urea soluble fraction increased with DNA demethylation, especially in the ALS patients. We added these data in Fig. 7b-e and the following sentences in the *Results* in the main text: “We also examined the TDP-43 protein levels in the motor cortex of 11 controls and 5 ALS patients whose tissue was used in the DNA methylation analysis (Fig. 7b). In the RIPA-soluble

fraction, there was no association between the DNA methylation of 3'UTR CpGs 10–15 and the TDP-43 levels (Fig. 7b, c). However, in the RIPA-insoluble urea-soluble fraction, the full-length TDP-43 protein levels (43 kDa) were inversely correlated with the percentage of DNA methylation of 3'UTR CpGs 10–15 (Fig. 7b, d). The TDP-43 C-terminal fragment (CTF) in the urea-soluble fraction also showed a similar trend with a significant correlation in the analysis of the ALS group only (Fig. 7b, e)."

Fig.7

Comment 3-1:

For the analysis of ALS patients, is it possible to include the data from other methylation study, e.g., Zhang et al., *Acta Neuropathologica*, 2020 (doi: 10.1007/s00401-020-02131-z)?

Response 3-1:

Since the previously reported ALS-related DNA methylation studies did not include the DNA methylation status in the *TARDBP* 3'UTR, we cannot integrate these studies with our data. However, based on this suggestion, we carefully considered how the concepts in other reports of ALS-related DNA methylation can be applied to our study. Then, we found a significant inverse correlation between an accelerated DNA methylation age and the age of onset in the ALS cases in our data set while referring to previous reports (Zhang, *Acta Neuropathol* 2017, Zhang, *Acta Neuropathol* 2020). We added these new data in Fig. 8 and the following sentences in the *Results* in the main text: " we calculated the acceleration of the DNA methylation age in the ALS cases after estimating the epigenetic age based on the DNA methylation of 3'UTR CpG 10 in the control motor cortex (Fig. 8a). As a result, the acceleration of the DNA methylation age was significantly inversely correlated with the onset age of ALS (Fig. 8b). No correlation between DNA methylation age acceleration and disease duration was observed (Fig. 8c)". We described the method used for the calculation of DNA methylation age acceleration in this study in the Figure legend of Fig. 8a as follows: "Epigenetic age' is estimated based on the regression coefficient calculated from the chronological age and the DNA methylation percentages at *TARDBP* 3'UTR CpG 10 in the control motor cortex (11 controls). The difference between the

epigenetic age and the linear regression model is defined as ‘DNA methylation age acceleration’ in the ALS cases.

Fig.8

Comment 3-2:

Similar to the previous points, could qRT-PCR/immunoblots for TDP-43 for these ALS samples be performed?

Response 3-2:

According to your recommendation, similar to the control samples, we performed a western blot analysis in addition to droplet digital PCR using the motor cortex of ALS patients.

First, we already show an inverse correlation between DNA methylation and the *TARDBP* mRNA levels in the motor cortex of control patients without brain disease. A similar relationship was not evident in the ALS cases. “Then, we investigated the association between the DNA methylation status of the 3'UTR CpGs and *TARDBP* expression in the motor cortex. The expression level of *TARDBP* mRNA tended to increase with DNA demethylation of 3'UTR CpGs 10–15 (Fig. 7a). In particular, the analysis of only the control group without brain disease showed a significant correlation (Fig. 7a).” There are several possible reasons why the results were not as hypothesized in the ALS motor cortex. In ALS-affected tissues, the expression of localization of several RNA-binding proteins that may affect RNA metabolism are disturbed. In addition, neuronal loss or gliosis may affect the expression levels of *TARDBP* mRNA measured in tissues. Therefore, the effect of the DNA methylation of the *TARDBP* 3'UTR may be ambiguous in ALS. However, in the present study, there was not enough evidence to definitively state these effects. Rather, it is possible that the power of detection was insufficient due to the small number of cases. Therefore, we rewrote the fifth paragraph of the *discussion* as follows: “it may be difficult to properly evaluate the motor cortex derived from ALS patients, who exhibit advanced neuronal loss and gliosis with whole tissues. In the future, methylation

analyses at the single-cell level should be performed to examine the methylation changes in each cell type” and “Finally, an analysis of a larger number of autopsy brains from patients without brain disease and ALS patients of varying ages should validate our conclusions in the future.”

In this revision, we performed a western blot analysis using tissue blocks (11 controls and 5 ALS patients) that were retained after being used for the DNA methylation analysis. As a result, we found that the demethylation of *TARDBP* 3'UTR CpGs 10-15 increased the amount of TDP-43 (full-length TDP-43) in the RIPA-insoluble urea-soluble fractions in the human motor cortex, although there was no correlation with the amount of TDP-43 in the RIPA-soluble fractions. Moreover, we showed that the C-terminal fragment (CTF) of TDP-43 in the RIPA-insoluble urea soluble fraction increased with DNA demethylation, especially in ALS patients. We added these data in Fig. 7b-e and the following sentences in the *Results* in the main text: "We also examined the TDP-43 protein levels in the motor cortex of 11 controls and 5 ALS patients whose tissue was used in the DNA methylation analysis (Fig. 7b). In the RIPA-soluble fraction, there was no association between the DNA methylation of 3'UTR CpGs 10–15 and the TDP-43 levels (Fig. 7b, c). However, in the RIPA-insoluble urea-soluble fraction, the full-length TDP-43 protein levels (43 kDa) were inversely correlated with the percentage of DNA methylation of 3'UTR CpGs 10–15 (Fig. 7b, d). The TDP-43 C-terminal fragment (CTF) in the urea-soluble fraction also showed a similar trend with a significant correlation in the analysis of the ALS group only (Fig. 7b, e). "

Fig.7

Other minor comments:

Comment 1:

Please elaborate on the gRNA(-) TET1 construct. Is this with “negative” gRNA or no gRNA, etc?

Response 1:

We changed the label 'gRNA (-)' to 'no gRNA' in Fig. 2c-e, 2i and Supplementary Fig. 1a, 1c,

1d.

Comment 2:

Could the author elaborate/rephrase line 88-90? It is not obvious what the authors are trying to say.

Response 2:

We agree that the sentence was confusing. We corrected this sentence to "These results indicate that TET1 affects each CpG on the allele as a cluster rather than stochastically."

Comment 3:

Although the numbers of human samples in Figure 5 were stated in the figure legend, would it be possible to indicate the numbers in each panel of Figure 5? The referee's intention is to clearly display the numbers of human samples in each analysis.

Response 3:

According to this recommendation, we indicated the number of cases of brain regions and liver tissues from the ALS and control cases in Fig. 5a, b.

Comment 4:

There are typographical errors in Materials and Methods section:

- a. Line 210: "Dulbecco's Modified Eagle's Medium" should all start with a capital letter.*
- b. Line 218: "causes the loss of the sixth of the 15 CpG sites in the 3'UTR" is this supposed to be "causes the loss of one-sixth of the 15 CpG sites in the 3'UTR"*
- c. Line 235 and 242: "AMpure" should be AMPure*
- d. Line 238-239: "D 7XX" and "D 5XX" should be "D7xx" and "D5xx" respectively*
- e. Figure 6e: "midle age" should be "middle age"*
- f. Supplementary Table 2:*
 - i. "targetting" should be "targeting"*
 - ii. "drolet" should be "droplet"*

Response 4:

Thank you for noting these issues. We corrected these errors. Regarding b. Line 218, we changed "causes the loss of the sixth of the 15 CpG sites in the 3'UTR" to "causes the loss of CpG 6 in the 3'UTR" to avoid confusion.

REVIEWERS' COMMENTS:

Changes in response to Reviewers 1 and 2 have been signed off.

Reviewer #3 (Remarks to the Author):

In this revised manuscript the authors have made a commendable effort to address my previous concerns. The new data on protein and mRNA analysis from Figure 7 and 8 in particular substantially strengthen the paper. I believe it is essentially suitable for Communications Biology in its current form with only one minor issue:

While it is quite clear that TDP-43 mRNA expression (from ALS patients) showed a clear correlation with DNA methylation status (Figure 7a), the TDP-43 protein does not appear to show significant difference based on the immunoblots (Figure 7b). The authors did argue the inverse correlation of TDP-43 protein with DNA methylation status (Figure 7c-e). Thus, it may be a bit misleading to state that "TDP-43 expression" is increased, as TDP-43 level per se did not show apparent difference between control and ALS patients. The authors may need to specify that TDP-43 "mRNA" expression is increased in both abstract and corresponding text. In addition, it may be good to include a "Ponceau-S stained" blots for the RIPA-insoluble urea-soluble fraction to show proper loading for Figure 7b.